# Cytoplasmic and Nuclear Functions of cIAP1

**DOI:** 10.3390/biom12020322

**Published:** 2022-02-17

**Authors:** Aymeric Zadoroznyj, Laurence Dubrez

**Affiliations:** 1Institut National de la Santé et de la Recherche Médicale (Inserm) Inserm LNC UMR1231, LabEx LipSTIC, 21000 Dijon, France; aymeric_zadoroznyj@etu.u-bourgogne.fr; 2Center de Recherche LNC UMR1231, University of Burgundy, 21000 Dijon, France

**Keywords:** IAPs, signaling pathways, innate immunity, ubiquitination, TNFα, NF-κB, cell migration, E2F1

## Abstract

Cellular inhibitor of apoptosis 1 (cIAP1) is a cell signaling regulator of the IAP family. Through its E3-ubiquitine ligase activity, it has the ability to activate intracellular signaling pathways, modify signal transduction pathways by changing protein-protein interaction networks, and stop signal transduction by promoting the degradation of critical components of signaling pathways. Thus, cIAP1 appears to be a potent determinant of the response of cells, enabling their rapid adaptation to changing environmental conditions or intra- or extracellular stresses. It is expressed in almost all tissues, found in the cytoplasm, membrane and/or nucleus of cells. cIAP1 regulates innate immunity by controlling signaling pathways mediated by tumor necrosis factor receptor superfamily (TNFRs), some cytokine receptors and pattern recognition-receptors (PRRs). Although less documented, cIAP1 has also been involved in the regulation of cell migration and in the control of transcriptional programs.

## 1. Introduction

IAPs (Inhibitors of Apoptosis) form a family of proteins highly conserved during evolution. The named “IAP” was chosen by Lois Miller’s teams, who described a new class of proteins encoded by the *Cydia pomonella granulosis* virus (CpGV) genome and were able to protect infected insect cells from apoptosis to allow viral spread [1]. Since then, based on sequence homologies, IAP homologs have been identified in insects, yeasts, nematodes, fish and mammals. However, although they are able to inhibit or delay cell death when overexpressed, the main cellular function for most of them is not an inhibition of apoptosis. Among the eight described mammal homologs, XIAP (X-linked IAP) [2], cIAP1 (cellular IAP1), cIAP2, ML-IAP (Melanoma IAP) [3] and ILP-2 (IAP-like protein 2) [4] are enzymes of the ubiquitination reaction involved in proteostasis and the regulation of the assembly of intracellular signaling platforms.

XIAP has the greatest ability to inhibit apoptosis by directly interacting with initiator and effector caspases and blocking their activity [5,6,7]. Upon apoptotic stimuli, XIAP is neutralized by Smac (Second mitochondria derived activator of caspase), which is contained into the mitochondrial intermembrane space and released into the cytosol during the early phase of apoptotic intrinsic pathway [8,9,10]. Based on these observations, Smac mimetics have been developed in order to counteract the anti-apoptotic activity of IAPs in tumors. However, Smac as well as Smac mimetics are also able to bind and neutralize other IAPs such as cIAP1, cIAP2 and ML-IAP with high affinities [11].

cIAP1 and its paralogous cIAP2 are most studied for their ability to regulate innate immunity and inflammation [12]. By controlling the scaffolding and kinase activities of RIPK1, cIAPs dictate the response of cells to tumor necrosis factor receptor (TNFR) superfamily stimulation [13]. They have also been involved in the control of the inflammatory response mediated by pattern recognition (PRRs) and cytokine receptors [14]. In addition, cIAP1 can control intracellular signaling pathways that drive cell motility and migration, regulate cell cycle cell proliferation and transcriptional programs [15]. 

The expression of cIAP1, cIAP2 and XIAP is preferentially induced under stressful conditions such as hypoxia, endoplasmic reticular stress and DNA damage. Regulation can occur at a transcriptional level via HIF-1α, NF-κB or E2F1-dependent mechanisms [16,17] or at translational level due to the presence of an internal ribosome entry site (IRES)-dependent mechanism of translation initiation [18,19,20]. Moreover, the stability of cIAPs is regulated by heat shock proteins (HSPs) [21]. Thus, these pleiotropic proteins appeared to act as potent regulators of the adaptive response of cells to a changing environment or in response to environmental or intracellular stresses such as pathogen attack, hypoxic-ischemic injury or DNA damage [22]. Consistent with their role in regulating homeostasis, dysregulations of cIAPs have been observed in cancer, neurodegenerative disorders and inflammatory diseases. Moreover, the oncogenic properties of cIAP1 were clearly demonstrated in mouse models of hepatocarcinoma, osteosarcoma and breast cancer [23,24,25]. This review aims to analyse the role of cIAP1 and, by comparison, cIAP2 and XIAP in maintaining cellular homeostasis.

## 2. cIAP1 Structure and Molecular Function

The IAP family is defined by its structural feature, i.e., the presence of at least one conserved protein domain named BIR (Baculoviral IAP repeat). cIAP1, cIAP2 and XIAP own three copies of BIRs (Figure 1). These domains have approximatively 70–80 amino-acids organized into three short β-strands and 4–5 α-helices forming a hydrophobic groove with protein-protein interacting properties [26]. The BIR1 of cIAP1/2 binds the signaling adaptor tumor necrosis factor receptor (TNFR)-associated factor 2 (TRAF2) (Figure 1) which regulates the stability, localization and activity of the concerned IAPs and which acts as an intermediate for their recruitment into TNFR-associated signaling complex [27,28,29]. The BIR2 and 3 of cIAPs and XIAP have the particularity of having a deep hydrophobic pocket, which allows the specific anchoring of a conserved tetrapeptide motif called IBM (IAP binding motif). The best characterized IBM-containing proteins are critical regulators of apoptosis Smac/diablo and HtrA2. cIAPs can also bind some caspases, the DNA damage response and cell cycle regulators chk1, eRF3/GSTP releasing factor and the kinase NIK in an IBM-dependent manner [15].

The second domain shared by cIAP1, cIAP2 and XIAP is the conserved Ring that gives them their molecular function. It is the widespread active domain found in E3-ligases of the ubiquitination reaction [30]. This is a 3-step enzymatic reaction that catalyzes the covalent binding of molecules or chains of ubiquitins of different topologies to protein substrates. This post-translational modification modifies the stability, localization, activity or the recruitment of intracellular proteins into signaling platforms, depending of the type of ubiquitin chains conjugated. Ubiquitination uses E1-activating, E2-conjugating and E3-ligase enzymes sequentially. It is generally admitted that the E3 is responsible for the recruitment of substrate proteins whereas the E2 determines the type of ubiquitination. IAPs bind, via the Ring, ubiquitin-charged E2-conjugating enzyme and catalyze the transfer of ubiquitin moieties from the E2 to the protein substrate, specifically recruited thanks to their BIR domains [15]. Like many Ring-containing E3-ligases, IAPs are active in a dimeric form [31,32]. The binding of ligand promotes their conformational change leading to activation [32]. Engagement of Smac mimetic to the BIR3-cIAP1 induces the activating dimerization and auto-ubiquitination of cIAP1 leading to its rapid degradation (within 15 min) [33]. We observed that cIAP1 and TRAF2 need each other to perform their respective activities. Increasing evidences suggests that they form an E3-ubiquitin ligase complex (Figure 1) in which cIAP1 functions as the E3-enzyme while TRAF2 serves as an adaptor for bring cIAP1 in close proximity to the substrates [34,35]. TRAF2 is also a potent regulator of cIAP1 stability [36,37]. In some situation, TRAF3 that directly binds TRAF2 takes part in the complex, serving as the substrate binding component [34,35,38,39]. In cytokine receptor-mediated signaling pathways, TRAF2, which also harbors a Ring domain (Figure 1) can function as an E3-ligase able to promote K63-linked ubiquitination and activation of cIAP1 [34].

In addition to the BIRs and Ring, cIAPs and XIAP harbor a UBA (ubiquitin- associated) domain whose function is to bind ubiquitins [40,41] (Figure 1). It has been involved in regulating cIAPs-mediated ubiquitination. It participates in the specific recruitment of ubiquitin-charged E2 [42] and therefore in determining the type of ubiquitination [43]. cIAP’UBA has also been involved in the recruitment of cIAPs into signaling platforms and in the binding to TRAF2 [41,43]. Moreover, cIAP1 and cIAP2 have a CARD domain that regulates their activating dimerization and enzymatic activity [44]. At least two functional NES sequences located in the BIR2-BIR3 linker region and in the CARD [45,46] were detected in the cIAP1 sequence.

## 3. Tissue Expression and Subcellular Localisation of cIAP1, cIAP2 and XIAP in Healthy and Tumor Cells

As documented in the human protein atlas (Human Protein Atlas proteinatlas.org) [47,48], cIAP1 is expressed in almost all tissues and cell types tested without specificity (Figure 2, Table 1). In comparison, cIAP2 is absent or less abundant in most tissues, except in the small intestine, kidney and lymphoid tissue. cIAP2 is highly expressed in subsets of immune cells, including B-cells.

cIAP1 has been found in the cytoplasm/membrane and/or nuclear compartments. In bone marrow hematopoietic cells, ovarian follicle cells, pancreas glandular cells, squamous epithelial cells of oral mucosa and cervix, hippocampus glial cells, lung alveolar cells and testis Leydig cells [49], it was exclusively detected in the nucleus (Table 1). In a work published in 2004, we demonstrated that cIAP1 is expressed in the nucleus of hematopoietic stem cells [50], and that its translocation into the cytoplasm is necessary for their differentiation into macrophages or dendritic cells [21,45,50]. Such nuclear export has also been observed during epithelial differentiation [21,50]. 

The subcellular localization of cIAP2 is much less documented in the literature, probably because of the low specificity of the available antibodies. The human protein atlas indicates a nuclear expression of cIAP2 in hematopoietic stem cells of bone marrow, spleen and lymph node cells, squamous epithelial cells of the vagina, cervix and oral mucosa, glandular cells from the stomach and salivary gland, and cells from the breast and urinary bladder (Table 1). On the other hand, XIAP has been found only in the cytoplasm/membrane compartment (Table 1). However, a nuclear translocation of XIAP has been observed in cells of the cortical region of the brain of neonatal rat exposed to hypoxic-ischemic brain injury. In the nuclei, XIAP interacts with XAF1 (XIAP-associated factor 1) [51]. The decreased cytoplasmic content of XIAP has been associated with enhanced caspase 3 activity and neuronal death [52]. XIAP has also been found in the nucleus of breast carcinoma cells [53].

Gene expression profiling interactive analysis (GEPIA) [54] revealed that *cIAP1* expression tends to be overexpressed in 11 out of 31 tumors selected in the cancer genome atlas (TCGA), which is significant for diffuse large B-cell lymphoma (DLBC), glioblastoma multiforme (GBM) and thymoma (THYM) (Figure 3). Nevertheless, it appears to be significantly correlated with overall survival only in lung adenocarcinoma. Conversely, *cIAP1* appeared downregulated in testicular germ cell (TGCT) and uterine cancers (UCEC) (UCS). cIAP1 and cIAP2-encoding genes (named *BIRC2 and BIRC3*) are very closely located on chromosome locus 11q22.2, a region found amplified (11q21 amplicon) in human medulloblastoma, hepatic, breast, pancreatic, cervical, lung, oral squamous cell and esophageal carcinoma [55]. Conversely, multiple myeloma is associated with inactivating mutations in genes involved in non-canonical NF-κB signaling pathways, which include *cIAP1* and/or *cIAP2*. At the protein level, cIAP1 expression does not emerge as a cancer prognostic factor in the cancers referenced in the human protein atlas. However, its nuclear expression has been correlated with overall survival, tumor stage or poor patient prognosis in cohorts of 70 cervical cancers [56], 102 bladder cancers [57] and 55 head and neck squamous cell carcinomas [58].

## 4. Cytoplasmic Functions of cIAP1

### 4.1. Role for cIAP1 in Regulating Innate Immunity

#### 4.1.1. Regulation of TNFα Signaling Pathways in Immune and Non-Immune Cells

The tumor necrosis factor alpha (TNFα) is the master regulator of tissue homeostasis by coordinating the inflammatory response and regulating the immune system (for review, see [59]). Dysregulated TNFR-signaling pathways or sustained production of TNFα has been involved in the pathogenesis of many chronic inflammatory diseases and anti-TNFα therapy has demonstrated efficiency in the treatment of severe forms of rheumatoid arthritis, Crohn’s disease, ulcerative colitis, psoriasis, psoriatic arthritis, ankylosing spondylitis and juvenile idiopathic arthritis. Conversely, neutralizing TNFα can also result in the onset of autoimmune disease supporting its pleiotropic functions in regulating the immune system [59,60]. It is produced within minutes of injury or stress, mainly by monocytes and macrophages, and it exerts its activity in transmembrane or soluble, secreted forms. TNFα is endowed with multiples functions depending on the cellular and environmental context. Its predominant activity is to trigger the production of pro-inflammatory cytokines and chemokines. It can also stimulate the survival and differentiation of immune cells, promote their recruitment to the site of damage, and enhance the adhesion of endothelial cells. Under specific conditions, survival signals can switch to cell death signals. For example, TNFα can help in killing infected cells in order to contain the infection and ensure tissue integrity; it takes part in the maintenance of peripheral immune tolerance by participating to the deletion of activated T-cells [61]; it can promote the death of irreversibly damaged cells in order to ensure tissue homeostasis [59]. 

TNFα is recognized by TNFR1 expressed in all human tissues and by TNFR2, whose expression is limited to immune cells, neurons, endothelial cells, cardiomyocytes, and osteoclast precursors. It is generally admitted that TNFR1 can trigger a strong inflammatory response and/or cell death, while TNFR2 induces cell death protection and a moderate inflammation. The response to TNFR1 stimulation is orchestrated by the presence of different checkpoints. The kinase RIP1 is critical in determining the inflammatory response or cell death. It is recruited into the surface receptor-associated intracellular complex via homotypic interaction thanks to the death-domain (DD) exhibited by both the receptor (intracellular side) and RIP1 [62]. In the receptor-associated signaling complex, so-called complex I, RIP1 acts as a scaffold for the recruitment of kinase complexes including TAK1/TAB2/TAB3 and IκB kinase (IKK) complex that promote MAPK and NF-kB-mediated transcriptional programs [63] (Figure 4). This scaffolding function is fully dependent on non-degradative poly-ubiquitination including K11, K63-linked, linear and hybrid-polyubiquitination [64,65]. On the other hand, thanks to its kinase activity, RIP1 can promote the assembly of a secondary cytoplasmic complexes including complex-II, ripoptosome and necrosome that result in apoptotic or necroptotic cell death [66] (Figure 4). Necroptosis is associated with a massive release of cytokines, chemokines and damage-associated molecular patterns (DAMPs) recognized by pattern recognition receptors (PRRs) that trigger the innate immune response [67,68]. The role of TNFα in chronic inflammatory diseases has been explained by its capacity to activate this immunogenic cell death [60].

cIAP1 takes part in this regulation. It constitutes an essential survival factor in intestinal epithelial cells, neutrophils, macrophages and activated T cells, allowing them to resist to TNFR1-mediated cell death when exposed to an acute inflammatory environment [69,70,71,72,73]. Depletion of cIAPs prevents TNFα-mediated NF-κB and MAPK activation and sensitizes cells to TNFα-mediated cell death [74,75,76,77]. In mice, deletion of cIAP1 as well as cIAP2 or XIAP did not lead to obvious phenotypic abnormalities. A moderate inflammation in lungs and intestines was observed in cIAP1^−/−^ KO mice [78]. However, double deletion of cIAP1 and cIAP2 or cIAP1 and XIAP in mice leads to embryonic lethality in TNFR1 and RIP1-dependent manner [75] and the specific depletion of cIAP1, -2 and XIAP in myeloid lineage or keratinocytes causes a severe local inflammation and TNFR1 or RIP1-dependent cell death [71,79,80]. By controlling the stability, scaffold function and kinase activity of RIP1, cIAPs have the ability to control the intensity and duration of the TNFR1-mediated inflammatory response: (i) they activate the scaffold function by promoting the conjugation of K11 and K63-linked poly-ubiquitin chains on components of complex I that include RIP1 [43,81,82,83]; (ii) they can stop the TNFR1-mediating signaling pathway by the promotion of ubiquitin-dependent degradation of RIP1 [43]; (iii) alternatively, cIAP-mediated ubiquitination of RIP1 represses its kinase activity necessary for the assembly of cell-death-mediated complexes-II [43] and then prevents TNF-mediated cytotoxicity and necroptosis-associated massive inflammation [84] (Figure 4). In addition to controlling the scaffold function, kinase activity and stability of RIP1, cIAP1 can regulate the TNFα-mediated NF-κB activating signalling pathway by the ubiquitination of NEMO/IKKγ (NF-κB essential modulator/IκB kinase-γ), the regulatory subunit of IKK complex [85].

TNFR2 plays a role in promoting the differentiation and stabilization of regulatory T cells, and mutation in TNFR2 has been involved in the pathogenesis of several autoimmune diseases [60]. In endothelial cells, it participates in tissue regeneration. Since the TNFR2 protein does not harbor DD (death- domain), it cannot recruit RIP1, but it can directly bind the molecular adaptors TRAF2 and TRAF3. TRAF2 recruits cIAP1 into the TNFR2-associated signaling complex. As observed in the TNFR1-associated signaling complex, cIAP1 can promote K63-linked polyubiquitinatin at the TNFR2-signaling complex [86], resulting in the recruitment and activation of kinase complexes that drive MAPK and canonical NF-κB. However, TNFR2 stimulation likely leads to cIAPs-dependent canonical NF-κB activation [86] (see below). 

#### 4.1.2. Regulation of the Non-Canonical NF-κB Signaling Pathway in Immune Cells, Osteoclasts and Endothelial Cells 

The best characterized substrate of the cIAP1/TRAF2 E3-ubiquitine ligase complex is NF-κB-inducing kinase (NIK), an essential mediator of the non-canonical NF-κB signaling pathway [34,87,88].

The non-canonical NF-κB signaling pathway is characterized by inducible processing of the p100 subunit in active p52 which, when heterodimerized with RelB, acts as a transcription factor. The processing of p100 is triggered following its phosphorylation by the IKKα homodimer, itself activated by NIK [89]. cIAP1 regulates the NF-kB alternative pathway by controlling the cellular content of NIK. In the resting condition, NIK is recruited to the cIAP1/TRAF2 complex via TRAF3. The complex is stabilized by direct binding of NIK with the BIR2 domain of cIAP1 in IBM-dependent manner [34,38,88]. cIAP1 promotes the ubiquitin-mediated proteasomal degradation of NIK, turning off the non-canonical NF-κB signaling pathway [34,87,88] (Figure 4). Stimulation of TNFR2, CD30, CD40, BAFF-R (B-cell-activating factor) or FN14 leads to the recruitment of TRAF2/TRAF3/cIAP1 complex to membrane-associated signaling complex [37,90,91,92,93]. TRAF2 induced cIAP1 activation via K63-linked ubiquitination. In turn, cIAP1 catalyzes K43-linked ubiquitination of TRAF2/3 and their degradation by the proteasome system, resulting in upregulation of NIK and activation of non-canonical NF-κB signaling [34].

Non-canonical NF-κB signaling is essential for the activation, survival and differentiation of immune cells such as B-cells, macrophages and dendritic cells. Deletion of cIAP1 and cIAP2 in mice maintained B-cells survival and maturation independent of BAFF-R stimulation [91], and can account for B-cell transformation [94,95,96]. We demonstrated that cIAP1-mediated degradation of TRAF2 is essential for the full activity of macrophages in response to CD40 stimulation [45]. IAP antagonists can also favor osteoclasts differentiation in a NIK-dependent manner, supporting the critical role of the non-canonical NF-κB signaling pathway in osteoclastogenesis [97].

#### 4.1.3. Regulation of PRR Signaling Pathways

The presence of pathogens in an organism is sensed by cell surface and intracellular receptors able to recognize a wide variety of pathogen-associated molecular patterns (PAMPs) and danger signals (DAMPs). Among them, the cell surface membrane TLR4, which recognizes bacteria lipopolysaccharides (LPS) can elicit distinct signaling pathways leading to either pro-inflammatory or interferon response. TLR4 engagement induces the recruitment of several cytoplasmic adaptor proteins thanks to the presence, in both the receptors and adaptors, of a homotypic interacting domain. The adaptor MyD88 (myeloid differentiation factor 88) has been involved in NF-κB and MAPK-dependent production of pro-inflammatory cytokines, whereas the adaptor TRIF (TIR-domain-containing adaptor-inducing IFN-b) is required for the IFN response. The cIAP1/TRAF2 E3-ubiquitine ligase complex is a potent determinant of the response to TLR4 stimulation. MyD-88 can directly recruit the adaptor TRAF3 which can bind the TRAF2/cIAP1 complex. In the MyD88-containing TLR4 complex (so-called Myddosome), the cIAP1/TRAF2 E3-ubiquitin ligase complex induces the ubiquitination and degradation of TRAF3, which results in the assembly of a secondary cytoplasmic signaling platform containing TRAF2/cIAP1, TAK1/TAB1–3 and IKK complexes leading to the activation of MAPK (Mitogen-activated protein kinases) and NF-κB (nuclear factor-kappa B)-signaling pathways [98,99,100] (Figure 4). Depletion of TRAF3 can also turn-off the IFN response that is involved in the TRAF6/TRAF3 complex.

In some situations, such as a sustained infection, the presence of pathogens resistant to inflammatory defense, or in some pathological conditions, TLR4, just like TLR3, which senses virus-derived nucleic acids, can also trigger RIP1-dependent cell death through a direct binding of RIP1 to the adaptor TRIF. cIAP1 constitutes a powerful survival factor in infected cells by preventing the assembly of ripoptosome and necrosome as explained above (4.1.1) [68,80].

Supporting the role of IAPs in controlling the strength and duration of the inflammatory response, Jin et al. showed that the cIAP1/TRAF2 complex can limit inflammation by promoting the ubiquitin-proteasome dependent degradation of c-Rel and IRF5 (interferon-responsive factor 5), two critical transcription factors involved in TLR-mediated NF-κB-dependent inflammatory and IFN response respectively. Depletion of TRAF2 in macrophages promoted colitis characterized by enhanced leukocyte infiltration in colon, mucosal damage and pro-inflammatory cytokines production in an animal model [35].

### 4.2. Role for cIAP1 in Cell Motility and Migration

Cell shape and cell motility are controlled by small GTPases of the Rho family. These proteins are critical regulators of the dynamic reorganization of the actin cytoskeleton [101,102,103]. They control cell architecture, focal adhesion complexes and local contraction by promoting the generation of stress fibers or membrane protrusions such as lamellopodia or filopodia [104]. They switch between a cytoplasmic, inactive GDP-bound state and a membrane-associated, active GTP-bound state, providing energy required for cytoskeleton rearrangement. RhoGTPase activation is mediated by guanine-nucleotide exchange factors (GEFs), which catalyze the transfer of GDP-bound to GTP-bound forms. Once activated, RhoGTPases are either recycled in inactive state by the action of GTPase-activation proteins (GAPs) or subjected to UPS-mediated degradation. The activation cycle of Rho GTPase is controlled by molecular chaperones such as guanine-nucleotide dissociation inhibitors (GDIs) which stabilize Rho GTPases in their cytosolic inactive state [104]. A relationship between IAPs and RhoGTPases was suggested in 2004 in a study showing that drosophila DIAP1 can interact with Rac1 and compensate for the migration defect triggered by the expression of a dominant negative form of this GTPase [105]. In mammals, in vitro assays have demonstrated that cIAP1, cIAP2 and XIAP are able to directly interact with the three most studied RhoGTPases [106,107,108,109] RhoA, Rac1 and cdc42, which promote lamellopodia, stress fibers or filopidia, respectively. In a study analysing the influence of cIAP1 on cell shape and migration, we demonstrated that cIAP1 can directly bind cdc42. It stabilizes cdc42 in its GDP-, inactive-state by promoting its association with its molecular chaperone RhoGDI. Deletion of cIAP1 deregulated the activation cycle of cdc42 by promoting its activation and then degradation [106]. Accordingly, cIAP1^−/−^ fibroblasts display an enhanced ability to migrate and exhibit filopodia. TNFα has the ability to induce cdc42 activation and actin reorganisation [102,103]. Upon TNFα stimulation, cIAP1 is recruited to the membrane receptor-associated complex, releasing cdc42 and promoting its activation [106] (Figure 4). The ubiquitination of cdc42 by cIAP1 has not been demonstrated; however, the ability of XIAP to ubiquitinate cdc42 and of XIAP and cIAP1 to ubiquitinate Rac1 has been observed [107,110]. Single or combined deletion of cIAP1, cIAP2 or XIAP differently affects cell shape, actin distribution and migratory capacity. They appear to have specific and distinct activity on each of the Rho proteins, suggesting that IAPs could regulate the spatiotemporal and sequential activation of Rho proteins [111]. Additional analysis will be required to decipher the regulation of the Rho proteins by IAPs.

## 5. Nuclear Functions of cIAP1

cIAP1 is a nuclear shuttling protein. Its nuclear expression has been correlated with the proliferative capacity of the cells. cIAP1 is excluded from the nucleus in cells undergoing differentiation [50]. Nuclear export is supported by the nuclear transport receptor Crm1 (chromosome region maintenance 1), which specifically recognizes leucine-rich nuclear export sequences (NES). Two NES were detected in the cIAP1 protein sequence. The first is located in the linked region between the BIR2 and the BIR3 [46] and the second in the CARD domain (Figure 1) [50]. These NES sequences are not conserved in cIAP2. Since cIAP1, as well as the cIAP2 and XIAP proteins do not contain NLS, an important issue is to understand the mechanisms of their nuclear accumulation. They could bind NLS-containing proteins. In the nucleus of B-cell, cIAP1 is complexed with TRAF2 and TRAF3, the latter containing a functional NLS in its TRAF-C domain [39]. Overexpression of the transcription cofactor Vestigial-like 4 (Vgl-4) have been shown to promote the nuclear translocation of cIAP2 [112], and the overexpression of the XIAP, cIAP1 and cIAP2-binding protein XAF1 (XIAP-associated factor 1) [113] triggered the nuclear retention of XIAP [51]. Intracellular protein movements are likely accompanied by post-translational modifications. Modifications of cIAP1 by ubiquitination, phosphorylation, S-nitrosylation and oxidation have been reported [114,115,116], but their roles in its subcellular distribution have not been investigated. Although cIAP1 has been found in cell nuclei in many tissues (Table 1), its nuclear functions remain poorly documented (Figure 5).

### 5.1. Regulation of Cytokinesis

In 2005, Samuel et al. showed that overexpression of cIAP1 increased the proportion of cells in the G2-M phase of the cell cycle and of polyploid cells, suggesting that cIAP1 could affect chromosome segregation. cIAP1 was observed in the midbody structures at the telophase where it colocalizes with Survivin. The binding of XIAP with survivin has also been reported [113]. Survivin is the smallest IAP member involved in chromosome segregation and cytokinesis [117]. Much work remains to be done to understand the role played by cIAP1 and XIAP in the regulation of cytokinesis.

### 5.2. Regulation of Transcriptional Program

We demonstrated that cIAP1 engages with chromatin. Different transcription factors are ubiquitination substrates of cIAP1. In 2011, the research of nuclear partners of cIAP1 revealed its binding with the transcription factor E2F1 [118]. This involves the cIAP1-BIR3 domain [17]. We demonstrated that cIAP1 and E2F1 are recruited together to the promoter of E2F-target genes [118]. Nuclear cIAP1 can promote K11- and K63-linked ubiquitination of E2F1 [17] and stabilize its protein expression. cIAP1-mediated K63-ubiquitination at Lysine 161/164 residues of E2F1 is required for its accumulation and transcriptional activation in the S phase of the cell cycle and in response to DNA damage [119]. Deletion of cIAP1 completely abrogated the binding of E2F1 onto DNA [17], suggesting that ubiquitin chains could act as a signal for the recruitment of the transcription factor to DNA. However, the underlying molecular mechanisms are not known. Of interest, the Lysine 161/164 residues are located in the DNA-binding domain of E2F1. The activation of the hypoxia inducible factor HIF1 that is responsible for adapting the transcriptional program in response to hypoxia is also controlled by K63-linked ubiquitination. In 2017, Park et al. demonstrated that XIAP can promote this modification, which results in the nuclear accumulation of the HIF1α subunit and the expression of the HIF1-responsive gene [120]. The ability of cIAP2 to stimulate the transcriptional program via non-degrading ubiquitination has also been reported. In 2014, Harikumar et al. demonstrated that IL-1 stimulation triggers the cIAP2-mediated K63-linked ubiquitination of IRF1-(interferon-regulatory factor1) resulting in activation and subsequent expression of IRF1-target genes. IL-1 induced the assembly of a signaling complex containing cIAP2, the adaptor TRAF6, the sphingosine kinase SphK1 and IRF-1. In the complex, SphK1 activation induced the local production of the bioactive lipid S1P (Shingosine-1-Phosphate) which acts as a cofactor for cIAP2 E3 ligase activity [121]. The nuclear expression of cIAP2 and XIAP does not seem required for the modification of IRF1 or HIF1α.

As mentioned above, cIAP1/TRAF2 E3-ubiquitine ligase complex is able to promote ubiquitination and degradation of the transcription factors c-Rel and IRF5 [35] and also the cAMP response element binding protein CREB [39]. The degradation of CREB in B-cells occurs in the nucleus and involves TRAF3, which bridges CREB to the E3-ubiquitine ligase complex [39]. Interestingly, CD40L stimulation in neurons has been observed to induce the translocation of TRAF2/TRAF3 complex into the nucleus where it can bind the NF-kB promoter element and act as a transcriptional regulator [122]. The presence of cIAPs in the complex was not analyzed.

cIAP1 can also indirectly modulate the activity of transcription factors c-myc by promoting the UPS-mediated degradation of its repressor Mad1 (Max dimerization protein 1) [123]. The authors suggest that cIAP1 could cooperate with c-myc to drive tumorigenesis. Since Mad1 is exclusively expressed in the nucleus, this is consistent with the nuclear localization of cIAP1 in the nucleus of tumor cells. Recently, a novel strategy for specifically inhibiting the E3-ubiquitine ligase activity of cIAP1 has been developed. Contrarily to the Smac mimetics which, by inducing a short pulse of cIAP1 activation prior to its degradation, destabilize Mad1 and thus promote c-myc activation, these novel compounds promote c-myc degradation [124].

cIAP1 is also able to bind the transcriptional cofactors Vestigial-like 4 (Vgl-4) [112]; and cIAP1, cIAP2 and XIAP can bind and modulate the stability of C-terminal binding protein 2 (CtBP2) [125].

In the nucleus, XIAP was detected associated to TCF/Lef transcriptional complexes whose expression is controlled by β-catenin. β-catenin moves the Groucho (Gro)/TLE transcriptional repressor and recruits a coactivator complex to enable TCF/Lef expression. When bound to TCF/Lef, XIAP can promote the monoubiquitination of GRO/TLE, which facilitates its detachment from TCF/Lef and favors the activation of the transcriptional program [126].

### 5.3. Cell Cycle Regulation

In addition to E2F1 that is essential for G1-S phase transition, the cyclin-dependent kinase (CDK) inhibitor p21 is a potential substrate of cIAP1. p21 is well known for its ability to arrest cell cycle progression in G1/S and G2/M transition by inhibiting CDK4/6/cyclin D and CDK2/Cyclin E complexes, respectively. Its half-life is regulated by post-translational modifications that included phosphorylation, ubiquitination and neddylation [127]. cIAP1 and cIAP2 can interact with p21. Downregulation of cIAP1 but not cIAP2 upregulated p21; however, it did not modify the ubiquitination profile of p21 but it did modify that of neddylation. The capacity of cIAP1 to directly induce ubiquitination or neddylation of p21 was not determined [128]. The subcellular localization of the cIAP-p21 interaction was not determined, however, because p21 is likely expressed in the nucleus; the capacity of cIAP1 to regulate p21 is consistent with its nuclear localization.

### 5.4. DNA Damage Response

A shRNA-based screening of ubiquitination-associated genes involved in DNA repair highlighted BIRC2 (cIAP1-encoding gene) and BIRC3 (cIAP2-encoding gene), the downregulation of which modulated chk1 activation [129]. Chk1 is a well known DNA damage sensor, preventing progression of cells into the cell cycle by promoting G2 arrest. Its protein sequence contains an IBM located just upstream of the initiator Methionine [130]. The ability of XIAP and also cIAP1 to interact with chk1 has been confirmed [130]. In the presence of XAF1, XIAP can promote chk1 degradation [131]

A role for cIAP2 in DNA damage response was also shown by Nicholson et al. in 2017 [132]. cIAP2 can complex with MRE11, a nuclease involved in homologous recombination (HR) and microhomology-mediated end-joining (MMEJ) repair pathways. cIAP2 was identified as an E3-ubiquitine ligase able to regulate the MRE11 protein level in cells exposed to HDAC (Histone deacetylase) inhibitors [132].

## 6. Conclusions

cIAP1 mainly exerts its activity by controlling the cell fate of its protein partners. Thanks to their ability to promote the conjugation of ubiquitin chains of different types, they can modulate the stability, localization and/or the activity of intracellular proteins and can change the composition of signaling platforms by modifying the intermolecular binding affinities. Thus, IAPs have the ability to control the implementation of signaling pathways and their regulations in time and space. To date, more than 30 cIAP substrates have been identified (recently reviewed in [15]). A database search for proteins containing IBM-like sequences found many proteins with different cellular functions [133], greatly expanding the number of potential IAP-binding partners. The identified IAP substrates are involved in various cellular processes essential for maintaining cell homeostasis (innate immune response, DNA damage response, cell cycle regulation). For most of them, the type and site of ubiquitination have not been determined. However, this is an important issue to address since they determine the cellular fate of the substrate [43].

The ultimate function of IAPs is to allow cells to adapt to their changing environment, to help implement an appropriate response to combat endogenous or exogenous stress or microbial aggression, and to restore homeostasis. Although loss of cIAP1 in mice has been associated with locale inflammation in lung, intestines or skin [78,79], deletion or mutation of the BIRC2/3 gene has not been associated with chronic inflammatory disease but has with cancer development. More in-depth studies of the implication of cIAPs in these pathologies deserve to be carried out. Most studies have focused on analyzing the role of cIAPs in innate immunity and in regulating cell surface receptor signaling pathways. However, consistent with their nuclear expression in cells in many tissues, their functions in the nucleus, in particular as a transcriptional regulator, may have been underestimated.

The expression of cIAP1 is ubiquitous and its regulation mechanisms are still poorly understood. The last observations suggest that cIAP1 and TRAF2 require each other and form an E3-ubiquitin ligase complex. cIAP1 E3-ligase activity is stimulated by K63-linked ubiquitination that can be mediated by TRAF2 or TRAF6 [34,98]. The stability of cIAP1 can be controlled by phosphorylation [114], and its regulation by S-nitrosylation and oxidation processes have also been reported [115,116]. One important issue to address concerns the mechanisms of regulation of the subcellular localization of cIAP1.

Smac mimetics designed to block XIAP anti-apoptotic activity are also potent inhibitors of cIAP1 by promoting its proteasome-mediated degradation. They have been developed as anticancer agents. However, because of the ability of cIAP1 to regulate RIP1 activities, numerous preclinical studies are exploring their potential in the treatment of inflammatory and infectious diseases.

## Figures and Tables

**Figure 1 biomolecules-12-00322-f001:**
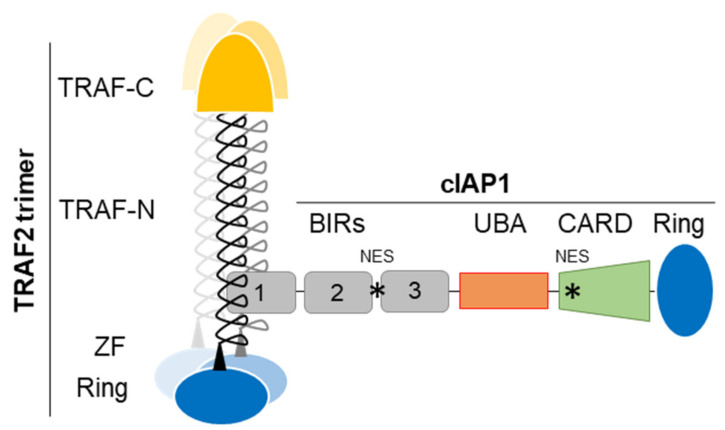
cIAP1-TRAF2 E3-Ubiquitin ligase complex, schematic representation. TRAF-N, TRAF-C, Zing finger (ZF), RING, BIRs (Baculoviral IAP repeat), UBA (ubiquitin-associated), CARD (caspase recruitment) domains and NES (Nuclear export signal) are represented. cIAP1-TRAF2 interaction involves the cIAP1-BIR1 domain and the N-terminal half of TRAF domain (TRAF-N). Trimeric TRAF2 is recruited to receptor thanks to their TRAF-C domain. Oligomerisation of TRAF2 is also required for the recruitment of downstream signaling molecules. Trimeric TRAF2 can bind one isolated BIR1.

**Figure 2 biomolecules-12-00322-f002:**
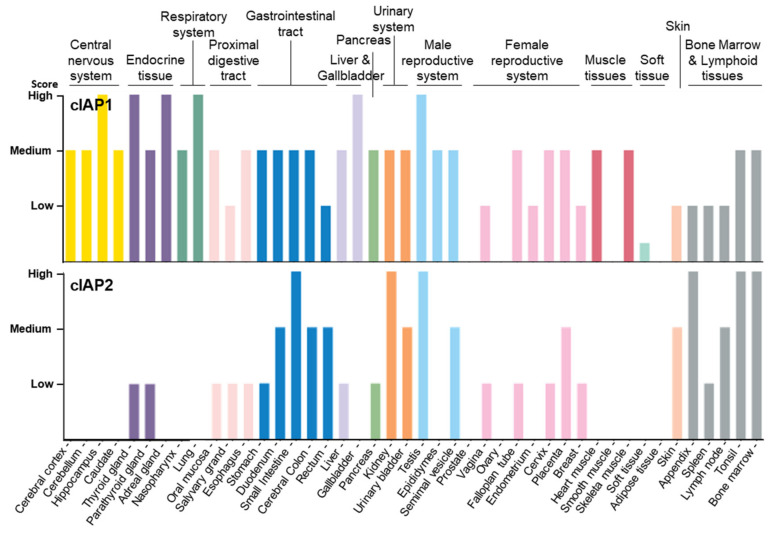
Tissue distribution of cIAP1 and cIAP2 proteins Data available from v21.0.proteinatlas.org (proteinatlas.org/ENSG00000110330-BIRC2/tissue, accessed on 23 January 2022; proteinatlas.org/ENSG00000023445-BIRC3/tissue, accessed on 23 January 2022).

**Figure 3 biomolecules-12-00322-f003:**
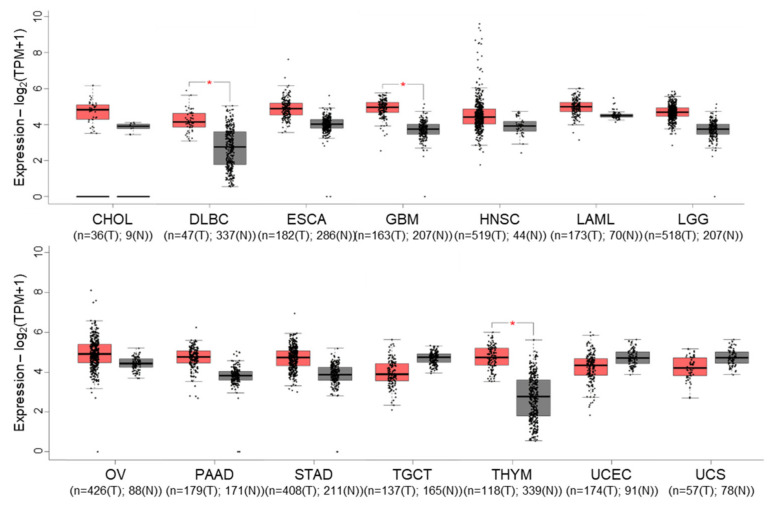
Gene expression profile interactive analysis (GEPIA) of *BIRC2* (cIAP1-encoding gene) in tumors (T, red)) and normal (N, grey) samples from the cancer genome atlas (TCGA) project. Only results showing a difference in *BIRC2* expression between cancer and normal samples are shown. CHOL: Cholangio carcinoma, DLBC: Lymphoid Neoplasm Diffuse Large B-cell Lymphoma, ESCA: Esophageal carcinoma, GBM: Glioblastoma multiforme, LAML: Acute Myeloid Leukemia, LGG: Brain Lower Grade Glioma, OV: Ovarian serous cystadenocarcinoma, PAAD: Pancreatic adenocarcinoma, STAD: Stomach adenocarcinoma, TGCT: Testicular Germ Cell Tumors, THYM: Testicular Germ Cell Tumors, UCEC: Uterine Corpus Endometrial Carcinoma, UCS: Uterine Carcinosarcoma. The method for differential gene expression analysis is one-way ANOVA, * *p* < 0.01.

**Figure 4 biomolecules-12-00322-f004:**
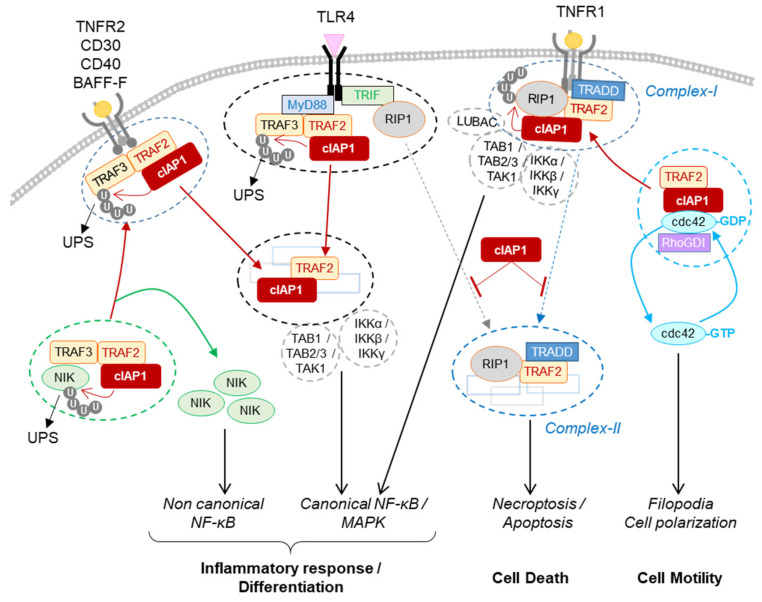
Regulation of signaling pathways by cIAP1. The cIAP1-TRAF2 E3-Ubiquitin ligase complex regulates the cellular content of NIK by mediating its ubiquitin-proteasome dependent degradation. The recruitment of cIAP1/TRAF2 to TNFR2, CD30, CD40 or BAFF-R releases NIK that in turn stimulates the non-canonical NF-κB signaling pathway. In the TLR4-associated signaling complex, cIAP1 induces the ubiquitination and degradation of TRAF3. cIAP1/TRAF2 forms a secondary cytoplasmic complex leading to NF-kB / MAPK activation. In TNFR1-associated complex, cIAP1 induces the ubiquitination of RIP1 and other components of the complex, resulting in the assembly of the signaling platform driving NF-κB and MAPK activation. cIAP1-mediated ubiquitination of RIP1 inhibits its kinase activity required for the assembly of cytoplasmic RIP-containing platforms leading to apoptotic or necrotic cell death. cIAP1 controls the cycle of activation of cdc42. The recruitment of cIAP1/TRAF2 to TNFR-associated signaling complex releases cdc42 for activation. BAFF-R: B-cell activating factor receptor; CD40-R: Cluster of differentiation 40 receptor, IKKα, β or γ: Inhibitor of κB kinase α, β or γ; LUBAC: linear ubiquitin chain assembly complex; Myd88: Myeloid differentiation primary response 88; NIK: NF-κB-inducing kinase; Rho-GDI: Rho-guanine-nucleotide dissociation inhibitors; TAB1, 2 or 3: transforming growth factor-activated kinase1-binding protein 1, 2, and 3; TAK1:tumor growth factor-β-activated kinase 1; TLR 4: toll-like receptor 4; TNFR2: tumor necrosis factor Receptor 2, TRADD: TNFR-associated death domain; TRIF: toll–interleukin 1 receptor domain-containing adaptor inducing IFN-β.

**Figure 5 biomolecules-12-00322-f005:**
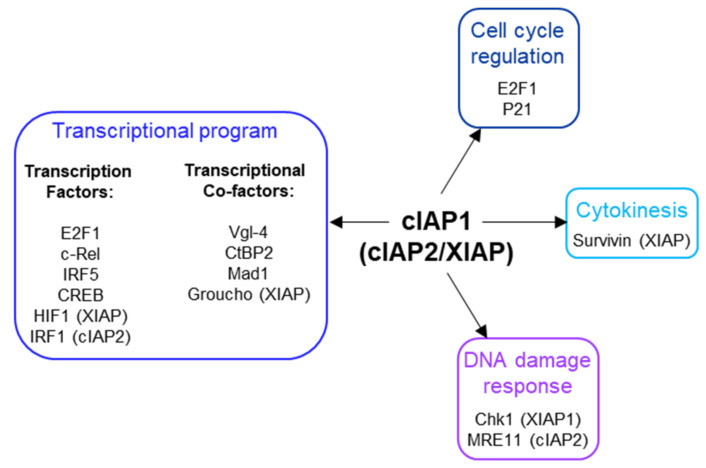
Nuclear partners of cIAP1, cIAP2 and XIAP.

**Table 1 biomolecules-12-00322-t001:** Tissue expression and subcellular localization of cIAP1, cIAP2 and XIAP. Data available from v21.0.proteinatlas.org (proteinatlas.org/ENSG00000110330-BIRC2/tissue, accessed on 23 January 2022; proteinatlas.org/ENSG00000023445-BIRC3/tissue; proteinatlas.org/ENSG00000101966-XIAP/tissue, accessed on 23 January 2022) [47,48].

Organ or System	Tissue	Cells	Protein Expression ^1^/Subcellular Localization ^2^		
			cIAP1	cIAP2	XIAP
**Adipose tissue**		Adipocytes	M/CMN	*nd*	L/CM
**Central nervous system**	Cerebellum	Cells in granular molecular layer purkinje cells	M/CMN M/CMN M/CMN	*nd* *nd* *nd*	L/CM *nd* L-M/CM
	Cerebral Cortex	Glial and neuronal cells	M/CM or N	*nd*	L-M/CM
	Hippocampus	Glial cells	H/N	*nd*	*nd*
		Neuronal cells	M/CM or N	*nd*	M/CM
	Caudate	Glial cells and neuronal cells	M/CM or N	*-*	-
**Endocrine**	Thyroid gland	Glandular cells	H/N or CM	L/CM	M-H/CM
**system**	Parathyroid Gland	Glandular cells	M/CM or CMN	L/CM	M/CM
	Adrenal Gland	Glandular cells	H/CMN	*nd*	M-H/CM
**Respiratory**	Nasopharynx	Respiratory epithelial cells	M/CM	*nd*	H/CM
**system**	Lung	Alveolar cells	H/N	*nd*	L/CM
**Gastrointestinal tract**	Oral mucosa Salivary Gland Esophagus Stomach	Squamous epithelial cells Glandular cells Squamous epithelial cells Glandular cells	M/N L/CM or CMN M/CM or CMN	L/N L/N L/N L/N	M/CM M-H/CM M/CM M/CM
	Duodenum and Small intestine and colon	Glandular cells	M/CM or CMN	*nd*	M-H/CM
	Rectum	Glandular cells	L/CM or CMN	*-*	M-H/CM
**Liver**		Cholangiocytes and hepatocytes	M/CM	L/CM	L-M/CM
**Gallbladder**		Glandular cells	H/CMN	*nd*	H/CM
**Pancreas**		Exocrine glandular cells	M/N	*nd*	L-M/CM
		Endocrine cells	M/N	L/CM	L-M/CM
**Urinary system**	Kidney	Glomeruli cells	L-M/CM or N	*nd*	L-M/CM
		Tubule cells	M/CM or CMN	-	M/CM
	Urinary Bladder	Urothelial cells	M/CMN	M/N	M/CM
**Female**	Vagina	Squamous epithelial cells	L/CM or N	L/N	L/CM
**reproductive**	Faloppian tube	Glandular cells	M/CM	-	M/CM
**system**	Endometrium	Glandular cells	L/CMN	*nd*	L-M-H/CM
	Cervix	Glandular cells	M/CM or CMN	*nd*	L-H/CM
		Squamous epithelial cells	L/N	L/N	L-M/CM
	Ovary	Ovarian stromal cells	L/CM or N	*nd*	L/CM
		Follicle cells	L/N	*nd*	*nd*
	Placenta	Trophoblastic cells	M/N or CMN	M/N	L-M/CM
	Breast	Glandular and myoepithelial cells	L/CM	L/N	M/CM
**Male reproductive**	Testis	Cells in seminiferous ducts Leydig cells	L/CM H/N	*nd* L	L/CM M-H/CM
**system**	Epididymis	Glandular cells	M/CM or CMN	*nd*	L-M/CM
	Seminal vesicle	Glandular cells	M/CMN	M/CM	M/CM
	Prostate	Glandular cells	L/CM or N	*nd*	L-M/CM
**Muscle**	Heart muscle	Cardiomyocytes	M/CM	*nd*	M-H/CM
**tissues**	Smooth muscle	Smooth muscle cells	L/CM	*nd*	L/CM
	Skeletal muscle	Myocytes	M/CM	*nd*	L-H/CM
**Skin**		Keratinocytes and melanocytes	L/CM	*nd*	L-M/CM
**Soft tissue**		Fibroblastes	L/CM	*nd*	L/CM
**Bone marrow &** **Lymphoid**	Appendix	Glandular cells and lymphoid tissue	L/CMN	*nd*	L-M-H/CM
**tissues**	Spleen	Cells in red pulp	L/CM or N	*nd*	M/CM
		Cells in white pulp	L/N	L/N	*nd*
	Lymph node	Germinal and non germinal center cells	L/CM or N	M/N	L-M/CM
	Tonsil	Germinal, non germinal center cells and squamous epithelial cells	M/N	H or M/N	L-M/CM
	Bone marrow	Hematopoietic cells	M/N	H/N	L-M/CM

^1^ L: low; M: medium, H: high; *nd*: not detected. ^2^ CM: cytoplasm/membrane; CMN: cytoplasm/membrane/nucleus; N: nucleus. Antibodies used for the immunohistochemical analyses: cIAP1: HPA005512, Sigma-Aldrich and CAB020661, Origene; cIAP2: HPA002317 Sigma-Aldrich; XIAP: HPA042428 Sigma-Aldrich and CAB009203, Santa Cruz Biotechnology.

## Data Availability

We use the human protein atlas (Human Protein Atlas proteinatlas.org), Gene expression profiling interactive analysis (GEPIA) (http://gepia.cancer-pku.cn/, accessed on 23 January 2022) and the cancer genome atlas (TCGA) to analyze the cellular and tissue distribution of IAPs.

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
