# Peer review of "Cytoplasmic and Nuclear Functions of cIAP1"

_biomolecules, 2022, doi:10.3390/biom12020322_

Round 1
Reviewer 1 Report
The review focuses on the cytoplasmic and nuclear functions of the cIAP1 protein. This is a very broad area which the authors cover very well. The majority of the text is focused on the cytoplasmic functions of cIAP1 as well as its molecular structure and mechanism of action. This area has been the major focus of the field for the last decade and thus warrants much attention in this review. The authors also included some data mined from the publicly available TCGA and human protein atlas projects, thus enriching the content of this review.
In relation, the section describing the nuclear functions of cIAP1 was rather short. I feel this section would benefit from deeper discussion about the potential functions and roles for cIAP1 in the nucleus. It would be interesting to discuss whether other IAPs have also been observed in the nucleus, if so under what conditions and to what end? Is there anything known about cross-talk between the cytoplasmic and nuclear roles of cIAP1 and the other IAPs? Since the nuclear functions of cIAP1 is in the title of this review, this section should be given more weight since this is rarely discussed in the literature or recent reviews on the IAP proteins.
In general, I find the text well written and organized and do not have any major criticisms regarding the language and style.
Author Response
The authors thank the referee for the suggestions that really improved our manuscript. We completed the section 3 “Tissue expression and subcellular localization of cIAP1 in healthy and tumor cells” and we also completed the Table 1 by indicating the location of cIAP2 and XIAP. The chapter 5 on nuclear functions of cIAP1 was reorganized. We included some sections discussing the role of cIAP1 and also cIAP2 and XIAP in cytokinesis, transcriptional regulation, cell cycle regulation and DNA damage response. We also discussed the mechanisms of nuclear translocation. We added a figure summarizing the nuclear functions of cIAP1 and also cIAP2 and XIAP.
Reviewer 2 Report
In this paper the authors study the cellular inhibitor of apoptosis 1 (cIAP1), a potent determinant of cellular response. Via the tumor necrosis factor receptor (TNFR), cIAP1 also regulates innate immunity. cIAP1 regulates innate immunity by controlling signaling pathways mediated by the tumor necrosis factor receptor (TNFR) superfamily. The authors conclude that cIAP1 also involved in the regulation of cell migration and in the control of the transcriptional program.
- In a recent interesting article, it was shown that some inflammatory functions also mediated by TNF, can be inhibited by the NLRP3 / Caspase-1 / IL -1β pathway. In this regard, to improve this article and make it more interesting to the readers of this important journal. I suggest the authors to read this relevant article below, incorporate their meaning and briefly report them in the discussion and in the list of references
- Protective effects of NLRP3 inhibitorMCC950 on sepsis-induced myocardial dysfunction. Li S, Guo Z, Zhang ZY.J Biol Regul Homeost Agents. 2021 Jan-Feb;35(1):141-150. doi: 10.23812/20-662-A.
- Furthermore, LINC00665 knockdown was observed to promote cell viability, while inhibiting cell apoptosis in PC12 cells. I suggest the authors to read this relevant article below, incorporate their meaning and briefly report them in the discussion and in the list of references
- LINC00665 knockdown protects against cerebral ischemia-reperfusion injury. Yan TS, Ma CH, Peng N, Li YE, Li QY, Wang H.J Biol Regul Homeost Agents. 2021 Jul 30;35(Spec Issue on Internal Medicine n.1). doi: 10.23812/21-SI1-8. Epub 2021 Jul 30.
Again, another interesting article reports that the protease inhibitor, oprozomib OPZ promotes phosphorylation of the JNK apoptosis signaling pathway, thus inducing apoptosis of cancer cells. I suggest the authors to also read this article below, incorporate their meaning and briefly report them in the discussion and in the list of references.
The new-generation proteasome inhibitor oprozomib increases the sensitivity of cervical cancer cells to cisplatin-induced apoptosis.
Li QZ, Sun LP, Shi HY, Chen Y, Shen H.J Biol Regul Homeost Agents. 2021 Mar-Apr;35(2):559-569. doi: 10.23812/20-504-A.
For these reasons I ask that the paper goes under minor revision.
Without these additions the paper cannot be published.
Author Response
I apologize for this but I did not understand the request. Suggested articles are not related to cIAP1 functions. The first one analyzed the efficiency of a selective NLRP3 inhibitor on sepsis-induced cardiac dysfunction. The second investigated the role of long intergenic non-protein coding RNA 665 (LINC00665) in cerebral ischemia-reperfusion in a rat model and in vitro in PC12 cells. They demonstrated that downregulation of LINC00665 could be beneficial in limiting cell death and attenuating damage caused by oxidative stress. Last, the third evaluated the anti-proliferative and pro-apoptotic effect of a combination of the protease inhibitor oprozomid (OPZ) with cisplatin. They demonstrated that OPZ inhibited cisplatin-induced NF-kB activation and enhanced JNK activation and apoptosis. There is no relationship with cIAP1.